# Past, Present and Future Climate Trends Under Varied Representative Concentration Pathways for a Sub-Humid Region in Uganda

**Anthony Egeru [1],\* , Bernard Barasa [2] , Josephine Nampijja [3], Aggrey Siya [4],
Moses Tenywa Makooma [5] and Mwanjalolo Gilbert Jackson Majaliwa [3]**

[1]  Department of Environmental Management, Makerere University, P.O. Box 7062 Kampala, Uganda
[2]  Department of Geography and Social Studies, Kyambogo University, P.O. Box 1 Kyambogo, Kampala, Uganda; barasagis@gmail.com
[3]  Department of Geography, Geoinformatics and Climatic Sciences, Makerere University, P.O. Box 7062 Kampala, Uganda; jnampijja09@gmail.com (J.N.); majaliwam@gmail.com (M.G.J.M.)
[4]  Department of Biosecurity, Ecosystems and Veterinary Public Health, Makerere University, P.O. Box 7062 Kampala, Uganda; tenywamakooma@yahoo.com
[5]  Department of Agricultural Production, Makerere University, P.O. Box 7062 Kampala, Uganda; tenywamakooma@yahoo.com
\*  Correspondence: egeru81@gmail.com; Tel.: +256-782-616-879

**Abstract:** Long-term trend analysis at local scale for rainfall and temperature is critical for detecting climate change patterns. This study analysed historical (1980–2009), near future (2010–2039), mid- (1940–2069) and end-century (2070–2099) rainfall and temperature over Karamoja sub-region. The Modern Era-Retrospective Analysis for Research and Applications (MERRA) daily climate data provided by the Agricultural Model Inter-comparison and Improvement Project (AgMIP) was used. The AgMIP delta method analysis protocol was used for an ensemble of 20 models under two representative concentration pathways (RCPs 4.5 and 8.5). Historical mean rainfall was 920.1 ± 118.9 mm and minimum, maximum and mean temperature were 16.8 ± 0.5 °C, 30.6 ± 0.4 °C and 32.0 ± 0.7 °C, respectively. Minimum temperature over the historical period significantly rose between 2000 and 2008. Near future rainfall varied by scenario with 1012.9 ± 146.3 mm and 997.5 ± 144.7 mm for RCP4.5 and RCP8.5 respectively; with a sharp rise predicted in 2017. In the mid-century, mean annual rainfall will be 1084.7 ± 137.4 mm and 1205.5 ± 164.9 mm under RCP4.5 and RCP8.5 respectively. The districts of Kaabong and Kotido are projected to experience low rainfall total under RCP4.5 (mid-century) and RCP8.5 (end-century). The minimum temperature is projected to increase by 1.8 °C (RCP4.5) and 2.1 °C (RCP8.5) in mid-century, and by 2.2 °C (RCP4.5) and 4.0 °C (RCP8.5) in end-century.

**Keywords:** climate change; trends analysis; temperature and precipitations; variability

## 1. Introduction

Drylands are important landscapes on a global scale, covering approximately 41% of the global land surface and supporting an estimated two billion people [1]. About 90% of those supported by the drylands live in developing countries, including those in Africa. Globally, drylands are classified into four sub-categories namely, hyper-arid (6.6%), arid (10.6%), semi-arid (15.2%), and sub-humid (8.7%) [2,3]. Recent studies [4–6] have shown that drylands, are on an onward increase in the recent decades. For example, between 1990 and 2004, drylands experience a 4% increase from the 1948 to 1962 period.

Dryland environments are inherently variable, characterised by stochastic rainfall patterns with a higher inter-annual and intra-annual variability [7]. Recent studies [8–10] are revealing a global intensification of inter-annual rainfall variability with an intensification of extreme weather events. According to [11], the long term global warming patterns tend to increase in semi-arid areas. As such, in warmer climates, droughts have tended to become longer in duration as well as intensity owing to increased evaporation and reduced precipitation [12,13]. Several studies [14,15] have shown increased drying tendencies in many semi-arid regions due to global warming as well as the drier regions becoming much drier. Consequently, projections into the dryland ecosystems have indicated that by 2100, the global drylands coverage will have increased by over 10% with the exception of India and some parts of northern tropical Africa which are predicted will become wetter [4]. According to [5], twenty global climate models predict that by 2100, under the condition of moderate (RCP4.5) and high-end scenario (RCP8.5) global drylands will increase by 11% and 23% relative to the 1961–1990 reference. Importantly, 78% of the dryland expansion will occur within the developing countries.

Dryland climates have generally received attention in terms of studies owing to the debilitating effects of extreme events especially droughts [16,17] and flash floods [18,19]. These studies to a large extent have focused on the historical to present climate patterns at a regional level. Whereas climate change projections have been undertaken, they have focused at coarse scale such as the sub-regional and global level. Yet, climate variability and change adaptation occurs at a local scale [20,21]. This calls for local scale level detailed information to be provided because planning for adaptation to climate change is often considered as a local imperative and regarded to be more effective if grounded on a solid evidence base that is considerate of relevant climate projections [22]. In this case, climate projections become critical given that planned adaptation to climate change, in particular involves the utilisation of information of the present and future climate situation so as to evaluate the present and planned practices, policies and infrastructure's suitability for adaptation [23].

Owing to the unwavering patterns of global warming, with significant effects and costs to humanity, the global community has relied on the coupled ocean-atmosphere general circulation models (AOGCM) to provide considerably accurate representation of the climate system [24]. These models have been used to represent the global and regional climates as well as providing relevant forecasts and projections over several time horizons. However, due to the fact that climate change adaptation needs are local-scale dependent, the spatial resolution required for local-level application needs is higher than that provided by the AOGCMs. Accordingly, downscaling techniques have been adopted to respond to such local level needs. Studies such as [25–27] over Uganda have applied these approaches. These albeit are minimal and such there are limited climate change projection studies undertaken at local scale to aid adaptation planning. This study provides a historical to future climate patterns in Karamoja sub-region in northeastern Uganda.

## 2. Materials and Methods

### 2.1. Study Area

Located in northeastern Uganda (Figure 1), Karamoja sub-region is one of the most misunderstood regions of Uganda. Inappropriate descriptions and references have through historical times been used in reference to this sub-region. The region is characterised by a dryland environment with semi-arid climate of sub-humid character whose stochastic rainfall is extremely variable, ranging between 500–800 mm annually [28]. The region's evapotranspiration is generally high. Temperatures are generally high throughout the year ranging from 15 °C to 18 °C for minimum and from 28 °C to 32.5 °C for maximum [29]. Owing to stochastic rainfall, the landscape is patched, characterised by savannah grasslands, woodlands, thickets and shrublands. This landscape is heterogeneous in nature with variation from east to west and north to south. Accordingly, the region's inhabitants are pastoral to agro-pastoral communities taking advantage of the region's heterogeneous landscape resources. Dependence on livestock as the primary source of livelihood has been declining since the disarmament

campaigns of the 2000s and the introduction of the alternative sources of livelihoods in particular promotion of sedentary crop production. Despite this heightened support towards crop production, climate variability remains a major constraints creating frequent water limitations leading to crop failure. The region has lately seen intermittent climate patterns laced with intense rainfall leading to flash floods and thereafter followed by intense droughts [30]. As the livelihood transitions continue to take place in the region, and in particular towards agro-pastoral production systems, a sound understanding of the present to future climate patterns is imperative to aid planned adaptation in the region.

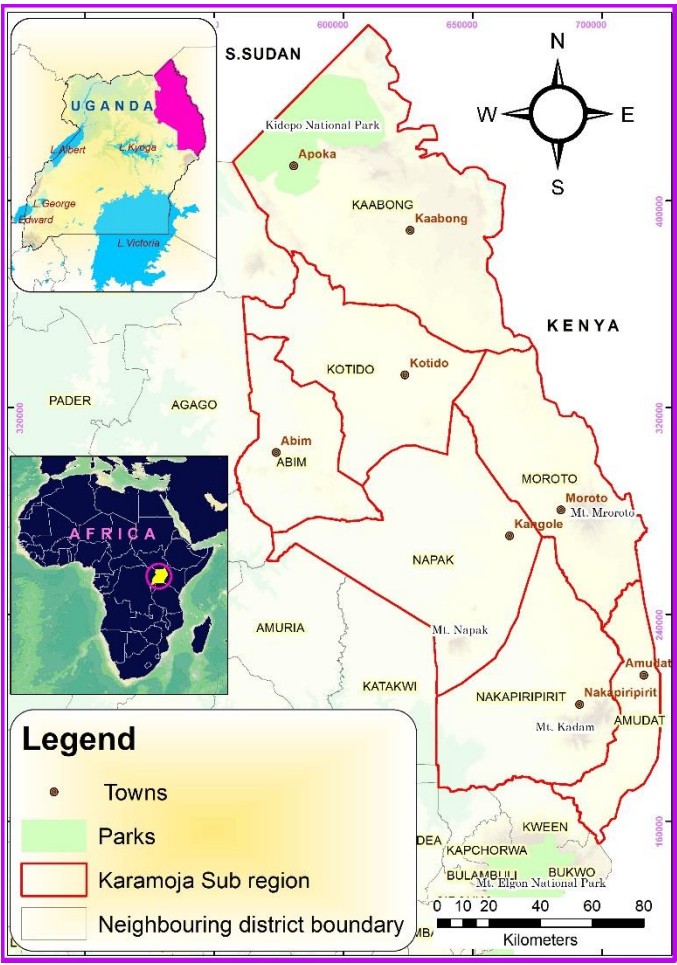

**Figure 1.** Location of Karamoja sub-region (Source: Authors).

## 2.2. Climate Data

In sub-Saharan Africa, climate data suffers from several discontinuities [31]. The situation is worse in areas that have had civil unrest such as the current study region because ground weather stations are often destroyed and where they are functional consistency in data collection is limited. To circumvent this challenge, we utilised Modern Era-Retrospective Analysis for Research and Applications (MERRA) daily climate data provided by the Agricultural Model Inter-comparison and Improvement Project (AgMIP). This provided historical (1980–2009) baseline climate time series data. MERRA is a product of the National Aeronautics and Space Administration (NASA)'s Global Modelling and Assimilation Office [32]. The data has under gone strict quality control mechanisms and complete descriptions of processes undertaken have been documented by [32]. Furthermore, observational assembly and advances made by the National Centers for Environmental Prediction (NCEP) reanalysis as well as the European Centre for Medium-Range Weather Forecasts (ECMWF) Reanalysis-ERA have considerably

benefited MERRA in improving data quality. A very high correlation (R2 = 0.81) between Climate Forecast System Reanalysis (CFSR) and MERRA climate data sets in the Karamoja sub-region was established. The application of MERRA data in scientific studies has tremendously gained ground with several researchers such as [33] having successfully utilised MERRA data.

## 2.3. Historical Trend Analysis

Descriptive statistics including minimum, maximum, mean and standard deviation were used to describe rainfall patterns in Karamoja sub-region. In order to determine the significance of the trend, the Mann–Kendall test was applied [34]. The Mann–Kendall is a non-parametric test approach that has considerably been used to determine the significance of the trend and in particular whether a trend exists in meteorological time series data [35]. In this study, we applied the Mann–Kendall test that has been widely used as a key analytical approach in trend analysis and as used by several researchers globally including in Ethiopia [36,37] and India [38]. The Mann–Kendall checked the null hypothesis of no trend versus the alternative hypothesis of the existence of increasing or decreasing trend in rainfall and temperature over the periods of analysis. The statistic (S) was defined as:

$$S = \sum_{i=1}^{N-1} \sum_{i=i+1}^{N} sgn(x_j - x_i), \tag{1}$$

where N is the number of data points; assuming that $(x_j - x_i) = \theta$, and the value of sign $(\theta)$ is computed as:

$$sgn\,(\theta) = (1\ if\ \theta > 0; 0\ if\ \theta > 0;\ -1\ if\ \theta > 0) \tag{2}$$

where this statistic represents the number of positive differences minus the number of negative differences for all differences considered. We conducted the test using the normal distribution because our sample was above the minimum ($N > 10$). Consequently, we computed the mean and variance as:

$$E(S) = 0, \tag{3}$$

$$var(S) = \frac{N(N-1)(2N+5) - \sum\limits_{k=1}^{n} t_k(t_k - 1)(2t_k + 5)}{18} \tag{4}$$

where n is the number of tied (zero difference between compared values) groups and $t_k$ is the number of data points in the $k^{th}$ tied group. We thus operationalized the standard normal deviate (Z-statistics) as:

$$Z = \begin{cases} \frac{S-1}{\sqrt{var(s)}}(if\,S > 0) \\ 0\,(if\,S = 0) \\ \frac{S+1}{\sqrt{var(s)}}(if\,S < 0) \end{cases} \tag{5}$$

where we held that if the computed value of $|Z| > z_{\alpha/2}$, the null hypothesis is rejected at $\alpha$ ($\alpha = 0.05$) level of significance in a two tailed test. We thus tested the null hypothesis at 95% confidence interval and in which case, a positive $Z_S$ value indicates increasing trend, otherwise it represents decreasing trend. At the 5% significance level, the null hypothesis of no trend is rejected if $|Zs| > 1.96$. The Mann–Kendall analysis was conducted using the MAKESENS application for trend calculation provided by the Finish Meteorological Institute.

## 2.4. Present and Future Climate Prediction for Karamoja Sub-Region

In projecting future climate of Karamoja, we utilised the Agricultural Inter-comparison and Improvement Project (AgMIP) delta method analysis protocol [39]. The delta method is based on the sum of interpolated anomalies to high resolution monthly climate surfaces. The method produces a smoothed (interpolated) surface of changes in climates (deltas or anomalies) and then applies

this interpolated surface to the baseline climate, taking into account the possible bias due to the difference in baselines [40]. Twenty models (ACCESS1-0, bcc-csm1-1, BNU-ESM, CanESM2, CCSM4, CESM1-BGC, CSIRO-Mk3-6-0, GFDL-ESM2G, GFDL-ESM2M, HadGEM2-CC, HadGEM2-ES, inmcm4, IPSL-CM5A-LR, IPSL-CM5A-MR, MIROC5, MIROC-ESM, MPI-ESM-LR, MPI-ESM-MR, MRI-CGCM3, and NorESM1-M) embedded in AgMIP protocol were used. Projection was conducted under three time slices; near-future (2010–2039), mid-century (2040–2069) and end-century (2070–2099) and two representative concentration pathways (RCP4.5 and RCP8.5). Representative Concentration Pathways are referred as pathways in order to emphasize that their primary purpose is to provide time-dependent projections of atmospheric greenhouse gas (GHG) concentrations. Further, the term pathway is meant to emphasize that it is not only a specific long-term concentration or radiative forcing outcome, such as a stabilization level that is of interest but also the trajectory that is taken over time to reach that outcome. They are representative in that they are one of several different scenarios that have similar radiative forcing and emissions characteristics [41]. The RCPs are part of radiative forcing pathways that have potential influence on the climate system in the 21st century [42]. The RCP4.5 describes the medium stabilization scenario without overshoot [43] while RCP8.5 describes rising radiative forcing pathway leading to very high emissions scenario [44]. In the analysis, we apply both concentration pathways in both time slices. Projection outputs were summarised into maximum, minimum and mean temperature. Annual rainfall on regional and spatial location basis was similarly computed using inverse distance weighted (IDW) interpolation in ArCGIS. The IDW is a geo-spatial statistical analysis approach that determines the cell values using a linearly weighted combination of a set of sample points. In order to test whether there is a significance difference in future rainfall and temperature conditions from the baseline conditions, a two-tailed t-test analysis was conducted on the data.

## 3. Results

### 3.1. Historical Rainfall and Temperature Trend (1980–2009) in Karamoja Sub-Region

Over the 30-year historical baseline period (1980–2009), average rainfall for the sub-region was $920.1 \pm 118.9$ mm per annum. This varied by district (Figure 2) with the lowest average rainfall received in Kotido and highest being received in Abim district (Table 1). Furthermore, long-term annual rainfall trend by district revealed periods of below annual average rainfall indicating drought years with Kotido, Moroto and Napak districts have a significant trend (Table 2) This pattern was also observed in terms of long rain and short rain periods (Figure A1). For the districts of Amudat, Kaabong, Kotido, Moroto, Napak and Nakapiripirit, a positive trend in long-term average was observed. This was particularly more pronounced after the year 2000 (Figure A1). The short rains trend is not pronounced because the sub-region is generally a unimodal region with trace rainfall during this period. The overall long-term rainfall trend of Karamoja during the 1980–2009 was not significant (Table 2).

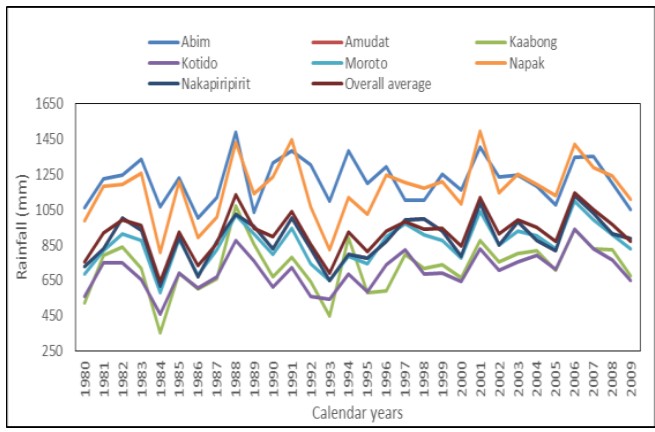

**Figure 2.** Historical rainfall trend for Karamoja sub-region.

**Table 1.** Summary of mean historical rainfall, minimum, maximum and average temperature (1980–2009) for Karamoja sub-region.

| District | Historical Rainfall Mean | Historical Minimum Temperature (°C) | Historical Maximum Temperature (°C) | Historical Average Temperature (°C) |
|---|---|---|---|---|
| Abim | 1216.8 ± 127.0 | 17.7 ± 0.4 | 31.7 ± 0.4 | 33.6 ± 0.6 |
| Amudat | 885.4 ± 125.6 | 16.2 ± 0.5 | 30.3 ± 0.4 | 31.3 ± 0.7 |
| Kaabong | 728.4 ± 147.4 | 16.2 ± 0.5 | 29.8 ± 0.4 | 31.1 ± 0.7 |
| Kotido | 701.6 ± 105.9 | 16.9 ± 0.5 | 30.2 ± 0.3 | 31.9 ± 0.6 |
| Moroto | 855.7 ± 117.9 | 16.2 ± 0.5 | 29.9 ± 0.4 | 31.2 ± 0.7 |
| Napak | 1167.5 ± 166.4 | 17.9 ± 0.5 | 31.7 ± 0.4 | 33.8 ± 0.8 |
| Nakapiripirit | 885.4 ± 125.6 | 16.2 ± 0.5 | 30.3 ± 0.4 | 31.3 ± 0.7 |
| Overall average | 920.1 ± 118.9 | 16.8 ± 0.5 | 30.6 ± 0.4 | 32.0 ± 0.7 |

**Table 2.** Zs values of Mann–Kendall trend test analysis rainfall, minimum and maximum temperature.

| District | Rainfall | Maximum Temperature | Minimum Temperature |
|---|---|---|---|
| | 1980–2009 | 1980–2009 | 1980–2009 |
| Abim | 0.39 | 1.14 | 2.71 * |
| Amudat | 1.43 | 2.28 * | 1.14 |
| Kaabong | 1.64 | 1.96 * | 3.64 * |
| Kotido | 2.14 * | 2.31 * | 4.07 * |
| Moroto | 1.89 | 2.99 * | 3.78 * |
| Napak | 1.53 | 3.71 * | 3.57 * |
| Nakapiripirit | 1.43 | 2.28 * | 3.78 * |

Figures with * denote statistically significant trend.

The average minimum, maximum and mean temperature were 16.8 ± 0.5 °C, 30.6 ± 0.4 °C and 32.0 ± 0.7 °C respectively (Table 1). The long-term trend revealed a significant rise in minimum temperature (Figure 3; Table 2). Meanwhile, maximum temperature varied with two high periods between 1986–1988 and 1999–2003 (Figure 4). Results of minimum temperature analysis for each month by district revealed that; March, April, May, September and November saw a progressive rise in minimum temperature in Abim district (Figure A1). Meanwhile, in the same district, a rise in maximum temperature was observed in the months of December, January, February, March, April (dry season months) and declining in the months of May, June, July, August, September and October (Supplementary material Table S2) For Amudat district, it was only January and September that showed a consistent rise in minimum temperature (Supplementary Material Table S3). Although variable, many of the results from the monthly analysis for most months took on a similar trend to that of Abim district with regard to minimum and maximum temperature (Supplementary Material Tables S4 and S5, Kaabong Tables S6 and S7, Kotido Tables S8 and S9, Moroto Tables S10 and S11, Napak Tables S12, S13 and Nakapiripirit Tables S1–S13). In the Supplementary Material Tables S1–S13, results of trends in the coefficient of variation for both inter and intra annual patterns, monthly variability and as well as the number of rainy days are provided.

*3.2. Near Future Rainfall and Temperature Trends in Karamoja Sub-Region*

The near future also regarded as the present day time slice (2010–2039) results revealed variability in rainfall trend with no distinct trend for both RCP4.5 and RCP8.5. However, there were observable differences in overall average rainfall in this period under the respective emissions scenarios. Results indicated that under RCP4.5, average rainfall will slightly be higher (1012.9 ± 146.3 mm) compared to that under RCP8.5 where average rainfall would be 997.5 ± 144.7 mm for this period (Table 3) During the present period, the average maximum temperature is 31.2 ± 0.4 (RCP4.5) and 31.4 ± 0.4 (RCP8.5) while minimum temperature is 17.7 ± 0.5 (RCP4.5) and 17.8 ± 0.5 (RCP8.5). Both maximum and minimum temperature under RCP4.5 and RCP8.5 minimally varied across the seven districts (Table 3).

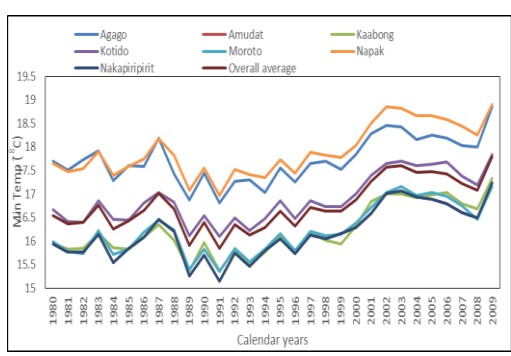

**Figure 3.** Historical minimum temperature for Karamoja sub-region.

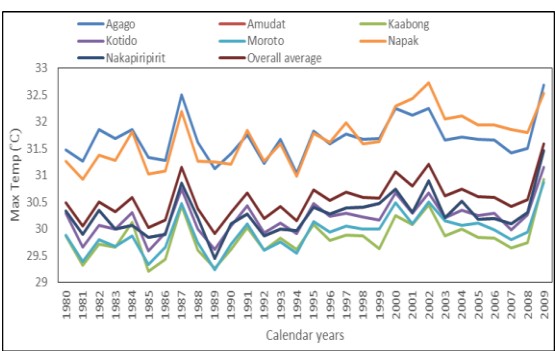

**Figure 4.** Historical trend of maximum temperature in Karamoja sub-region.

**Table 3.** Summary of maximum and minimum temperature and Rainfall under RCP4.5 and RCP8.5 in the Near future.

| District | Tmax RCP4.5 | Tmax RCP8.5 | Tmin RCP4.5 | Tmin RCP8.5 | Rainfall RCP4.5 | Rainfall RCP8.5 |
|---|---|---|---|---|---|---|
| Abim | 32.3 ± 0.4 | 32.5 ± 0.4 | 18.6 ± 0.5 | 18.7 ± 0.5 | 1346.1 ± 142.1 | 1312.8 ± 137.3 |
| Amudat | 31.0 ± 0.4 | 31.1 ± 0.4 | 17.1 ± 0.5 | 17.1 ± 0.5 | 972.5 ± 139.6 | 978.4 ± 142.3 |
| Kaabong | 30.5 ± 0.4 | 30.6 ± 0.4 | 17.1 ± 0.5 | 17.2 ± 0.5 | 794.7 ± 164.3 | 775.5 ± 161.0 |
| Kotido | 30.9 ± 0.3 | 31.0 ± 0.3 | 17.8 ± 0.5 | 17.9 ± 0.5 | 763.4 ± 116.7 | 746.5 ± 115.1 |
| Moroto | 30.6 ± 0.4 | 30.8 ± 0.4 | 17.1 ± 0.5 | 17.3 ± 0.5 | 950.1 ± 133.2 | 931.8 ± 131.4 |
| Nakapiripirit | 31.0 ± 0.4 | 31.1 ± 0.4 | 17.1 ± 0.5 | 17.1 ± 0.5 | 972.5 ± 139.6 | 978.4 ± 142.3 |
| Napak | 32.3 ± 0.5 | 32.6 ± 0.5 | 18.8 ± 0.5 | 19.0 ± 0.5 | 1286.7 ± 188.3 | 1258.9 ± 183.8 |
| **Overall average** | **31.2 ± 0.4** | **31.4 ± 0.4** | **17.7 ± 0.5** | **17.8 ± 0.5** | **1012.9 ± 146.3** | **997.5 ± 144.7** |

*3.3. Projected Rainfall in Mid and End Century in Karamoja Sub-Region*

Marked inter-annual variability in mid-century (2040–2069) and end century (2070–2099) will be expected but when compared to the 1980 baseline period, rainfall in the end century will be higher for both emissions scenarios. Disaggregated results showed that mean annual rainfall between time slices (mid-century 2040–2069 and end-century (2070–2099) will variedly continue to rise. Variability within and between time-slices as well as in different scenarios will generally remain high (Table 4). Spatially, projected total annual rainfall will generally be above 700 mm mark under both RCP4.5 and RCP8.5 scenarios in all locations. However, spatial and temporal variability within and between locations in all scenarios (Table 4; note the high standard deviations) will be expected (see Figures 5–8). In mid-century, Abim, Amudat, Nakapiripirit and Napak districts will be the only areas with a more pronounced presence of above 1000 mm rainfall total under all scenarios (Figures 5 and 6). Abim is the only area that will experience an overall rainfall decline under RCP4.5 in mid-century (Table 4) but return to a positive average during end-century (2070–2099). Notably, under all-time slices, projected total rainfall will be higher under RCP8.5 scenario (Table 4). With exception of Amudat under RCP8.5

end-century, the t-test results showed that all the other districts and the sub-region's rainfall showed a significant difference with the baseline historical rainfall in Karamoja (Table 4).

**Table 4.** Projected rainfall in Karamoja sub-region.

| Analysis Time Slice | Projected Mean Annual Rainfall (mm) | | Change in Rainfall (mm) | |
|---|---|---|---|---|
| Sub-Region's Average | RCP4.5 | RCP8.5 | RCP4.5 | RCP8.5 |
| Mid-century (2040–2069) | 1084.7 ± 137.4 * | 1190.6 ± 166.3 * | 180.8 ± 180.5 | 206.6 ± 229.7 |
| End-century (2070–2099) | 1107.5 ± 147.3 * | 1220.3 ± 165.1 * | 177.3 ± 203.7 | 210.0 ± 228.0 |
| Century period (2040–2099) | 1084.7 ± 137.4 * | 1205.5 ± 164.9 * | 180.8 ± 180.5 | 244.9 ± 226.1 |
| District level average | | | | |
| Mid-century (2040–2069) | | | | |
| Abim | 1401.9 ± 139.9 * | 1407.4 ± 143.2 * | −99.4 ± 223.2 | 33.6 ± 226.3 |
| Amudat | 1024.5 ± 149.3 * | 1046.6 ± 154.2 * | 212.5 ± 200.8 | 236.1 ± 206.2 |
| Kaabong | 829.6 ± 172.0 * | 1686.4 ± 350.2 * | 197.9 ± 235.5 | 428.3 ± 478.2 |
| Kotido | 799.1 ± 123.2 * | 800.9 ± 124.9 * | 113.1 ± 153.6 | 123.1 ± 156.6 |
| Moroto | 999.9 ± 141.8 * | 998.5 ± 143.9 * | 188.3 ± 184.9 | 202.3 ± 187.2 |
| Nakapiripirit | 1024.5 ± 149.3 * | 1046.6 ± 154.2 * | 212.5 ± 200.8 | 236.1 ± 206.2 |
| Napak | 1354.1 ± 196.9 * | 1348.0 ± 197.9 * | 174.5 ± 281.8 | 186.9 ± 282.5 |
| End-century (2070–2099) | | | | |
| Abim | 1463.4 ± 151.3 * | 1610.8 ± 163.7 * | 25.4 ± 238.2 | 39.6 ± 258.8 |
| Amudat | 1079.0 ± 158.2 * | 1248.9 ± 185.1 | 236.3 ± 210.3 | 293.1 ± 247.0 |
| Kaabong | 856.8 ± 177.6 * | 949.3 ± 201.4 * | 217.9 ± 242.4 | 261.3 ± 274.4 |
| Kotido | 827.7 ± 127.3 * | 924.9 ± 146.9 * | 126.9 ± 158.9 | 157.7 ± 185.2 |
| Moroto | 1037.5 ± 147.6 * | 1165.8 ± 170.0 * | 207.0 ± 191.7 | 251.7 ± 221.5 |
| Nakapiripirit | 1079.0 ± 158.2 * | 1248.6 ± 185.1 * | 236.3 ± 210.3 | 293.1 ± 247.0 |
| Napak | 1408.8 ± 206.2 * | 1563.6 ± 231.4 * | 192.1 ± 293.8 | 230.4 ± 328.0 |

* t-test result indicating a significance difference with the baseline (1980–2009) rainfall.

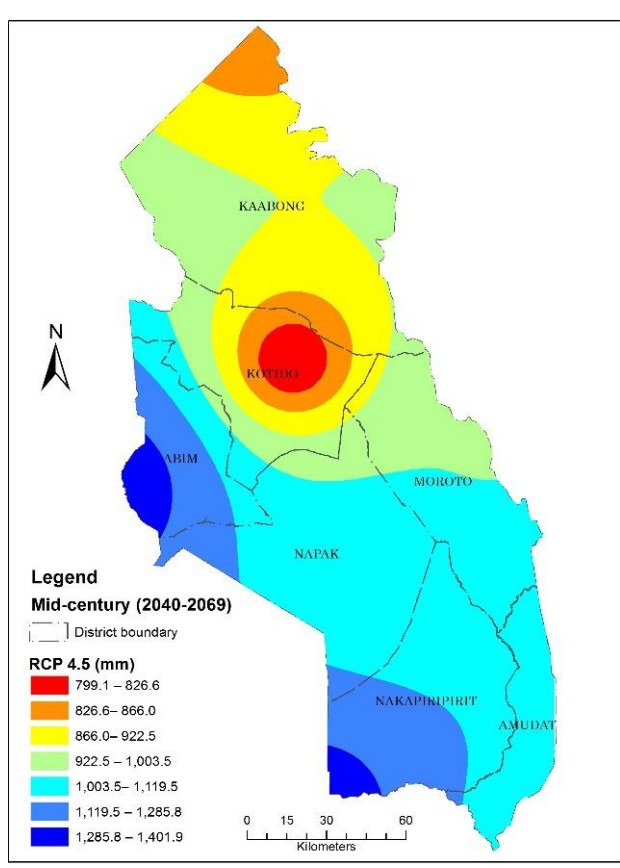

**Figure 5.** Rainfall distribution under RCP4.5 mid-century.

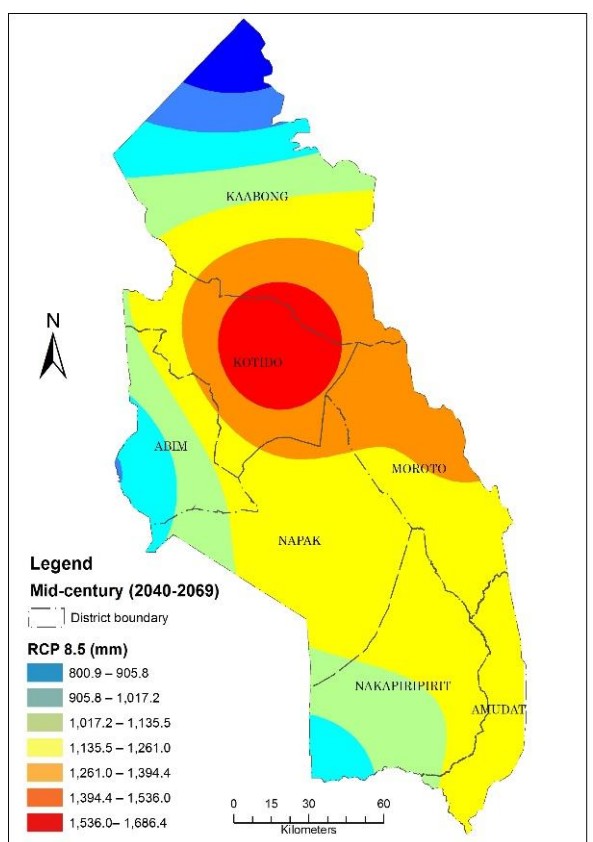

**Figure 6.** Rainfall distribution under RCP8.5 mid-century.

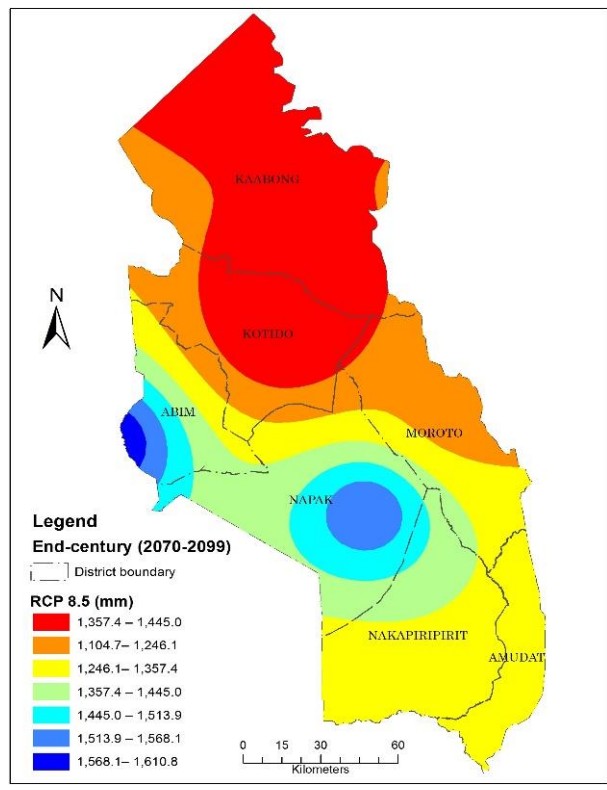

**Figure 7.** Rainfall distribution under RCP4.5 end-century.

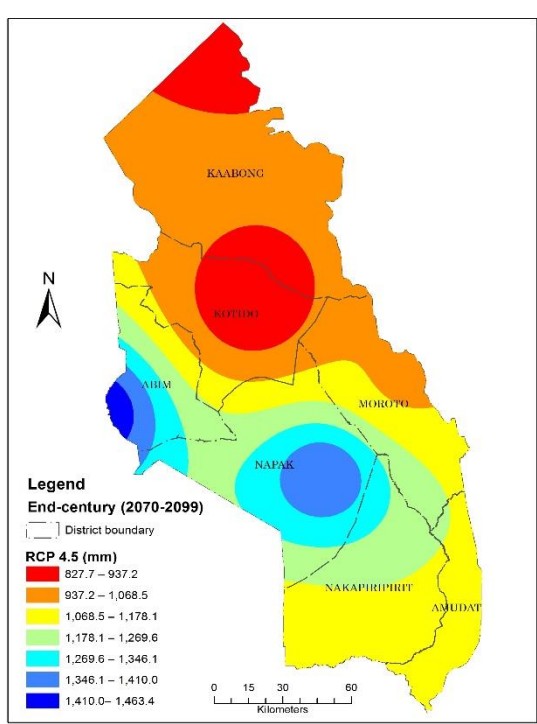

**Figure 8.** Rainfall distribution under RCP8.5 end-century.

## 3.4. Projected Temperature in Mid- and End-Century in Karamoja Sub-Region

All the 20 model ensemble showed a similar trend in projected maximum, minimum and mean temperature in respective time slices and emission scenarios. Under RCP4.5, maximum temperature will rise slightly in mid-century and end-century. On the other hand, maximum and minimum temperature under RCP8.5 scenario will continuously rise throughout the 21st century. Similarly, mean temperature will experience a progressive rise throughout the 21st century under both RCP4.5 and RCP8.5. Maximum, minimum and mean temperature will all be higher under the high emissions pathway-RCP8.5 (Table 5). At district level, Napak and Abim districts will generally have higher maximum, minimum and mean temperatures compared to Nakapiripirit, Kotido, Kaabong, Amudat and Moroto districts under all-time scales and emissions scenarios (Figures 9 and 10). The t-test results revealed that there was a significance difference in the mid-century and end-century from the baseline (1980–2009) temperature conditions (Table 5).

**Table 5.** Future projected temperatures in Karamoja sub-region.

| Analysis Time Slice | Projected Temperature (°C) | | Change in Temperature (°C) | |
|---|---|---|---|---|
| | RCP4.5 | RCP8.5 | RCP4.5 | RCP8.5 |
| Mid-century (2040–2069) | | | | |
|     Tmax | 31.9 ± 0.3 * | 32.3 ± 0.4 * | 0.8 ± 0.3 | 1.1 ± 0.4 |
|     Tmin | 18.5 ± 0.5 * | 18.9 ± 0.5 * | 1.2 ± 0.3 | 1.3 ± 0.4 |
|     Tmean | 25.2 ± 0.4 * | 25.6 ± 0.4 * | 0.9 ± 0.3 | 1.2 ± 0.3 |
| End-century (2070–2099) | | | | |
|     Tmax | 32.3 ± 0.4 * | 33.6 ± 0.5 * | 1.1 ± 0.4 | 1.3 ± 0.4 |
|     Tmin | 18.9 ± 0.5 * | 20.8 ± 0.5 * | 1.2 ± 0.4 | 0.2 ± 0.4 |
|     Tmean | 25.6 ± 0.4 * | 27.2 ± 0.4 * | 1.2 ± 0.3 | 0.7 ± 0.3 |
| Long term period (2040–2099) | | | | |
|     Tmax | 32.1 ± 0.4 * | 33.2 ± 0.7 * | 1.4 ± 0.4 | 1.4 ± 0.4 |
|     Tmin | 18.7 ± 0.5 * | 20.1 ± 1.0 * | 1.7 ± 0.3 | 3.0 ± 0.4 |
|     Tmean | 25.4 ± 0.4 * | 26.7 ± 0.9 * | 1.6 ± 0.3 | 2.2 ± 0.3 |

* t-test result indicating a significance difference with the baseline (1980–2009) temperature.

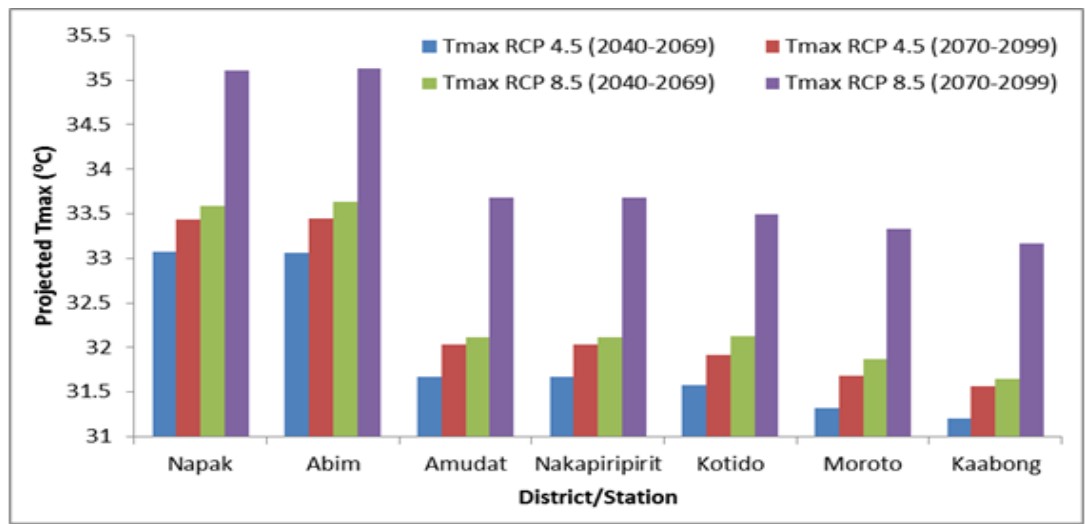

**Figure 9.** Projected maximum temperatures at district level in Karamoja.

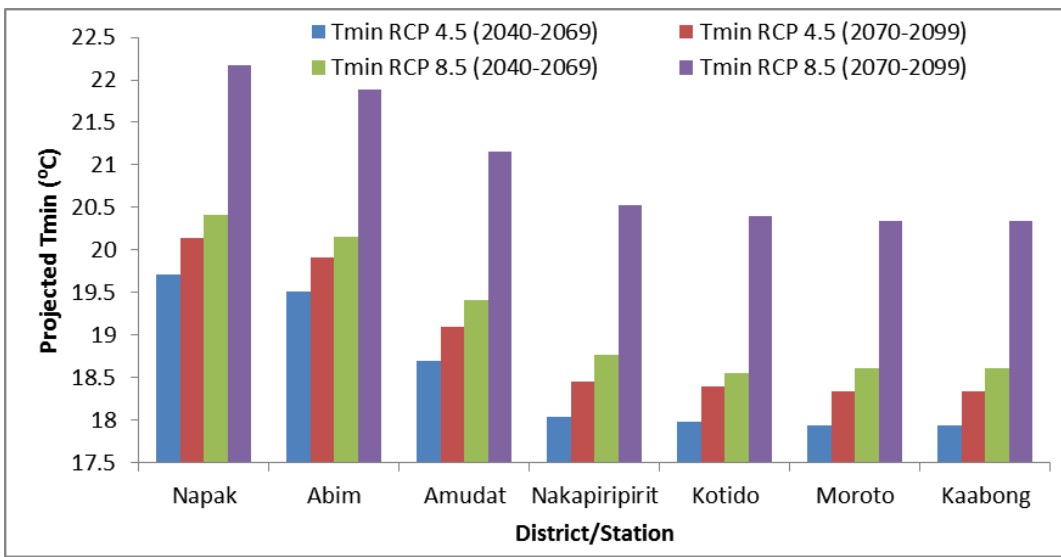

**Figure 10.** Projected minimum temperatures at district level in Karamoja.

When the baseline climate was compared to the projected climate states, results showed that there will be an increase in both maximum and minimum temperatures. At sub-region level, minimum temperature in the mid-century (2040–2069) will be expected to increase by 1.8 °C and 2.1 °C from the 1980–2009 baseline period under RCP4.5 and RCP8.5 respectively. The increase in minimum temperature from the baseline period is expected to continue, such that the end century will be much warmer than the mid-century and the baseline period. During the end century, minimum temperature is expected to increase by 4.0 °C under RCP8.5 emissions scenario. Similarly, maximum temperature will increase but the increase will be lower than that expected under minimum temperature (Table 6).

**Table 6.** Comparison of change in temperature from the baseline (1980–2009) period.

| Projected Change in Minimum Temperature (°C) from Baseline (1980–2009) Period | | | | |
|---|---|---|---|---|
| | RCP4.5 Mid | RCP8.5 Mid | RCP4.5 End | RCP8.5 End |
| Napak | 0.6 | 1.0 | 1.0 | 2.9 |
| Abim | 0.8 | 1.2 | 1.2 | 3.1 |
| Kotido | 1.6 | 2.0 | 2.0 | 3.9 |
| Moroto | 2.3 | 2.7 | 2.7 | 4.6 |
| Kaabong | 2.3 | 2.7 | 2.7 | 4.6 |
| Amudat | 2.4 | 2.8 | 2.8 | 4.7 |
| Nakapiripirit | 2.4 | 2.8 | 2.8 | 4.7 |
| Sub-region's average | 1.8 | 2.1 | 2.2 | 4.0 |
| **Projected Change in Maximum Temperature (°C) from Baseline (1980–2009) Period** | | | | |
| | RCP4.5 Mid | RCP8.5 Mid | RCP4.5 End | RCP8.5 End |
| Napak | 0.2 | 0.6 | 0.6 | 1.9 |
| Abim | 0.2 | 0.6 | 0.6 | 1.9 |
| Amudat | 1.6 | 2.0 | 2.0 | 3.3 |
| Nakapiripirit | 1.6 | 2.0 | 2.0 | 3.3 |
| Kotido | 1.7 | 2.1 | 2.1 | 3.4 |
| Moroto | 2.0 | 2.4 | 2.4 | 3.7 |
| Kaabong | 2.1 | 2.5 | 2.5 | 3.8 |
| Sub-region's average | 0.3 | 1.7 | 1.7 | 3.0 |

## 4. Discussion

### 4.1. Historical Trend in Rainfall and Temperature

Water remains a primary factor limiting and regulating the ability and functioning of ecosystems in dryland ecosystems. In water-limited environments such as the sub-humid regions, stochastic pulses of water influence plant life and species composition and structure that in turn influence water fluxes at larger scales [45]. This study has revealed inherent historical and long-term variability in rainfall patterns in Karamoja over a 30-year (1980–2009) period of analysis. Results of this study in the same period showed a rising total rainfall after the year 2000 further revealing the internal stochastic patterns in the climate of the sub-region. Earlier analysis by [30] over the sub-region had indicated that after 2000, the region was experiencing an increased wetness intensity with intermittent occurrence of flood and drought events. Such stochastic patterns introduce complex system dynamics that often tend to lead to nonlinear responses within these environments [46]. Whilst the rainfall was variable and showing a rising trend, minimum temperature in the sub-region was similarly rising. The months of March, April, May, September and November in particular had a distinct rising trend in minimum temperature. Rising minimum temperature within the tropics has been widely reported. In an analytical climate study covering 1971–2009 in India, the rise in minimum temperature has been observed with wide scale negative effects on agriculture especially reduction is yield per hectare [47] In essence, a rise in minimum temperature increases the overall mean temperature; this has implications on the ecological processes given changes in evapotranspiration that it triggers. This tends to lead to ecological droughts in such environments. Consequently, ecosystem's performance such as in vegetation productivity is affected, this destabilises key production resources-pasture and water availability at landscape level [48].

### 4.2. Present-Near Future Trend in Rainfall and Temperature

The global concentration of carbon dioxide in the atmosphere has presently (near future) passed the 400 parts per million (ppm) regarded by researchers as a threshold [49,50]. Results of this study show that present situation-near future based on the results of 20 model ensemble for both RCP4.5 and

RCP8.5 revealed a general warming up of the Karamoja sub-region. Minimum temperature over the region will continue to rise and this will continue influence the character and nature of the overall mean temperature over the region. Global rise in minimum, maximum and mean temperature has been particularly been reported with sharp increases being observed in the 2000-2010 decade in 138 climate record [51]. According to [25], Uganda in general will experience an increase in temperature during the near future 2010–2039. The results of this study are thus in agreement with the earlier findings of [28] that showed warmer years beginning from around 2000. Earlier, [52] had articulated that should carbon dioxide concentrations continue to rise, which the global climate science community has continued to observe [53,54] several locations on a global scale will experience a new and permanent heat regime over the next four decades.

Karamoja's rainfall in the near future will continue to be variable with observable low below average rainfall total in some years such as 2023, 2032 and 2034 expected. These years are thus expected to be drought years during this period. Drought analysis over Uganda shows that Uganda, experienced droughts in 2008, 2009, 2010 and 2011 [55,56] 2014 and 2016 [57]. These periods identified in these earlier studies correspond the results that this study has established as there was relatively low rainfall totals recorded in the historical rainfall period in 2007 and 2008 as well as in 2014 and 2016.

### 4.3. Mid- and End-Century Rainfall and Temperature Trends in Karamoja Sub-Region

Results of this study showed a consistency in rainfall trend for both RCP4.5 and RCP8.5 in the mid- and end-century periods. Despite this consistency in trend, overall total variability and inter-annual variability will be expected with a relatively lower total under RCP4.5 compared to RCP8.5 conditions. These results emerge against the assumption that continued global warming effect will lead to decreased rainfall total. It however, corroborates findings of [58] that conducted a projected climate analysis over the Horn of Africa and showed that there will be increased rainfall of over 100 mm by the end of the 21st century under RCP8.5 scenario. Indeed, the overall rainfall difference between the RCP4.5 and RCP8.5 scenarios is estimated at 121 mm in the end century. These results however contrast with those of [59] that have shown that the reductions in rainfall total is projected over the areas of Limpopo in South Africa. However, a possibility of overall increase in rainfall over the interior of South Africa is anticipated in the mid-century [60]; thus revealing the spatial connections associated with climate trends over locations. Considering the spatial extent of the sub-region, results revealed a spatiality effect with both scenarios in the end-century (2070–2099) period indicating northern Karamoja as a hot spot location of considerably reduced rainfall totals. According to [28], northern Karamoja and in particular Kotido district, suffers considerably more from droughts and prolonged dry spells (87%) compared to southern Karamoja areas such as Nakapiripirit district (50%). As such, the end century projections seem to indicate that this situation will continue to exist despite slight adjustments in mid-century under the RCP4.5 scenario.

Minimum, maximum and mean temperature will continue to rise under both RCP4.5 and RCP8.5 projections. Results also indicated that the rise will particularly be significant under RCP8.5 scenario indicating the effect of intensifying carbon dioxide emission concentrations in the atmosphere as well as that of other greenhouse gases [61]. The observed widening of the gap between RCP4.5 and RCP8.5 for minimum temperature could perhaps be attributed to the differences in peaking levels for the different emissions scenario. The RCP4.5 and RCP8.5 have been shown to have different peaking levels. For example, it has been observed that RCP4.5 will have carbon dioxide concentrations rising throughout the century clocking a maximum level of 540 ppm in 2100. On the other hand, the CO2 concentrations will progressively rise peaking at 935 ppm in 2100 [62]. It is these differences in carbon dioxide concentration levels that perhaps explain the observed differences in overall projected temperature increase of to 4.0 °C under RCP8.5 and to 2.2 °C under RCP4.5. This is perhaps expected, considering that RCP4.5 is considered as a scenario that stabilizes the radiative forcing after 2070 until 2100 [63]. However, the observed changes are also within the projected global range of change in temperature as observed by [64] in particular respect to the RCP8.5 scenario.

## 5. Conclusions

Past, present and future rainfall and temperature trends have been analysed for the Karamoja sub-region. Rainfall totals received in the sub-region when compared to other dryland areas in Eastern Africa confirm that the Karamoja is a sub-humid drylands belt. Second, within the sub-region, spatial variability in rainfall pertains as well as a strong element of inter-annual variability. These variability patterns predispose the region to complex systems cycles due to strong stochastic patterns. From the near future period (2010–2039), it is evident that minimum temperature will continue to progressively rise towards the mid-century period. Maximum temperature on the other hand will gradually rise during the period. Future projections (mid- and end-century) further revealed the continued rise in minimum, maximum and mean temperature. However, temperature trends will be projection scenario (RCP4.5, RCP8.5) dependent. There will be a continued presence of strong spatial and temporal rainfall variability over the sub-region. Northern Karamoja, and in particular the districts of Kaabong and Kotido, will become hotspot locations of reduced rainfall total under RCP8.5 in mid-century and under both RCP4.5 and RCP8.5 in end century. On the other hand, the districts of Abim and Napak in central to western Karamoja will remain better off in overall total rainfall. Based on these historical, near-, mid- and end-century trend analysis, application of these outputs to test the overall effects on the productivity of the ecosystem is recommended in order to better design climate smart adaptation options for the sub-region.

## 6. Patents

There are no patents associated with this work.

**Supplementary Materials:** The following are available online at http://www.mdpi.com/2225-1154/7/3/35/s1, File: Table S1 and S2. Abim Minimum and Maximum temperature, Table S3 and S4 Amudat minimum and maximum temperature, Table S5 and S6 Kaabong minimum and maximum temperature, Table S7 and S4.2 Kotido minimum and maximum temperature, Table S8 and S9 Moroto minimum and maximum temperature, Table S10 and S11 Nakapiripirit minimum and maximum temperature, and Table S12 and S13 Napak minimum and maximum temperature.

**Author Contributions:** Conceptualization, Methodology, A.E. and M.G.J.M.; Software, Validation, B.B.; M.T.M. and J.N.; Formal Analysis, A.E.; Investigation, A.E.; Data Curation, M.T.M., J.N. and A.S.; Writing-Original Draft Preparation, A.E.; Writing-Review and editing, M.G.J.M. and A.S.; Visualization, A.E.; Supervision, M.G.J.M.; Project Administration, A.E.; Funding Acquisition, A.E. and M.G.J.M.

**Funding:** This research was funded by Regional Universities Forum for Capacity Building in Agriculture (RUFORUM) grant number [RU/2012/DRRG/01/004] with support from the Carnegie Corporation of New York.

**Acknowledgments:** Regional Universities Forum for Capacity Building in Agriculture (RUFORUM) and Makerere University for the opportunity that was accorded to Anthony Egeru to pursue PhD studies that formed the basis for developing these sorts of articles.

**Conflicts of Interest:** The authors declare no conflict of interest.

## Appendix A. Annual, Long and Short Rainfall Trends 1980–2009

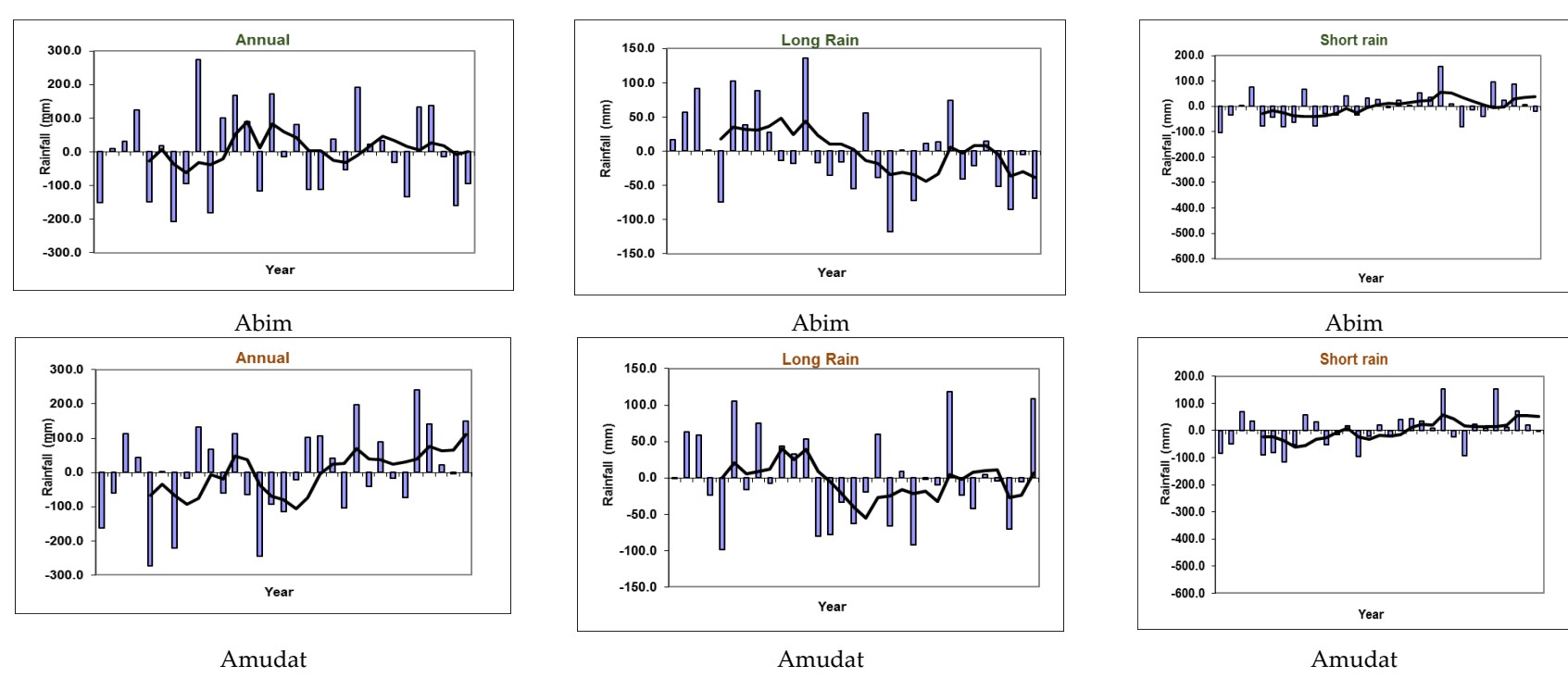

**Figure A1.** *Cont.*

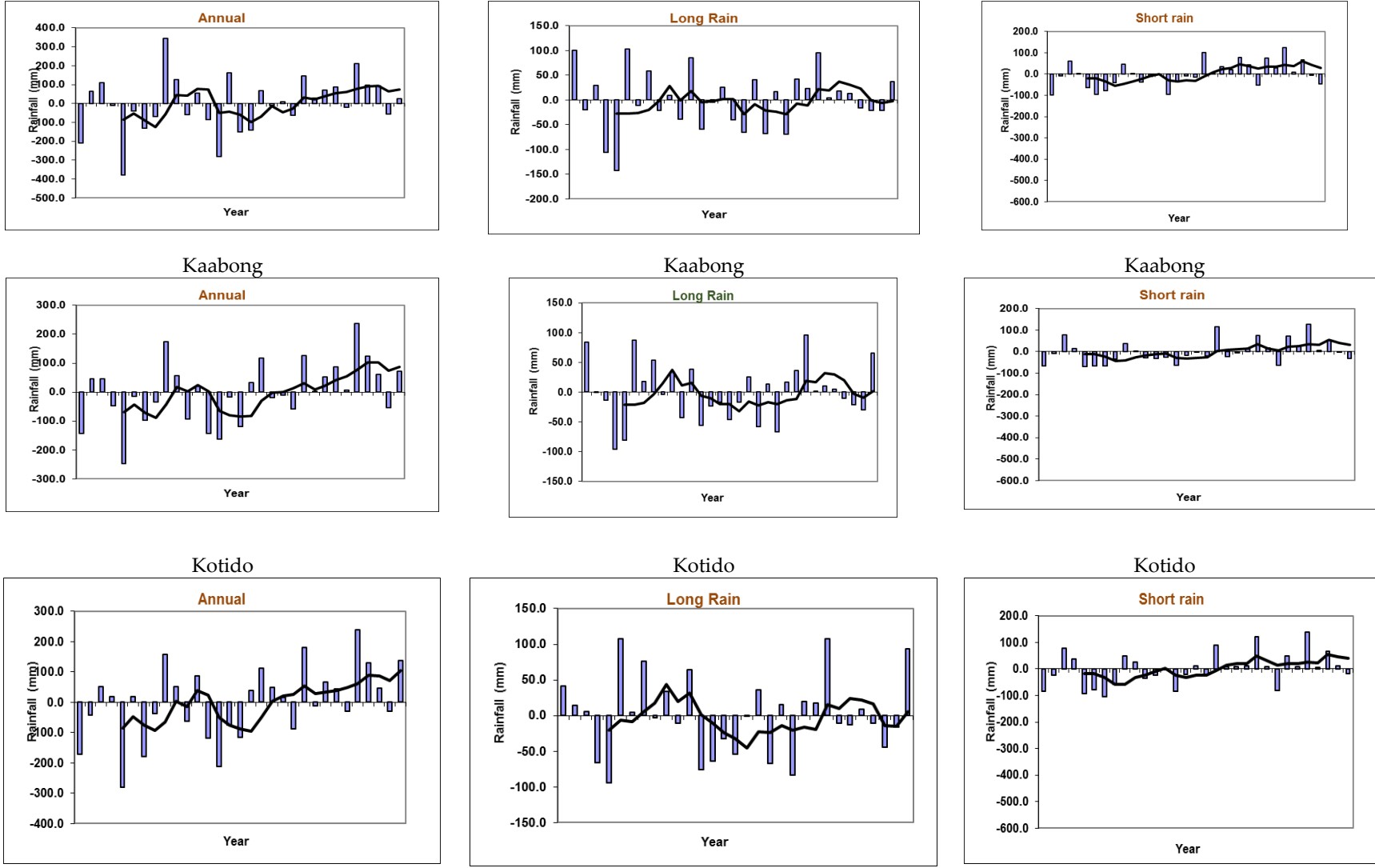

**Figure A1.** *Cont.*

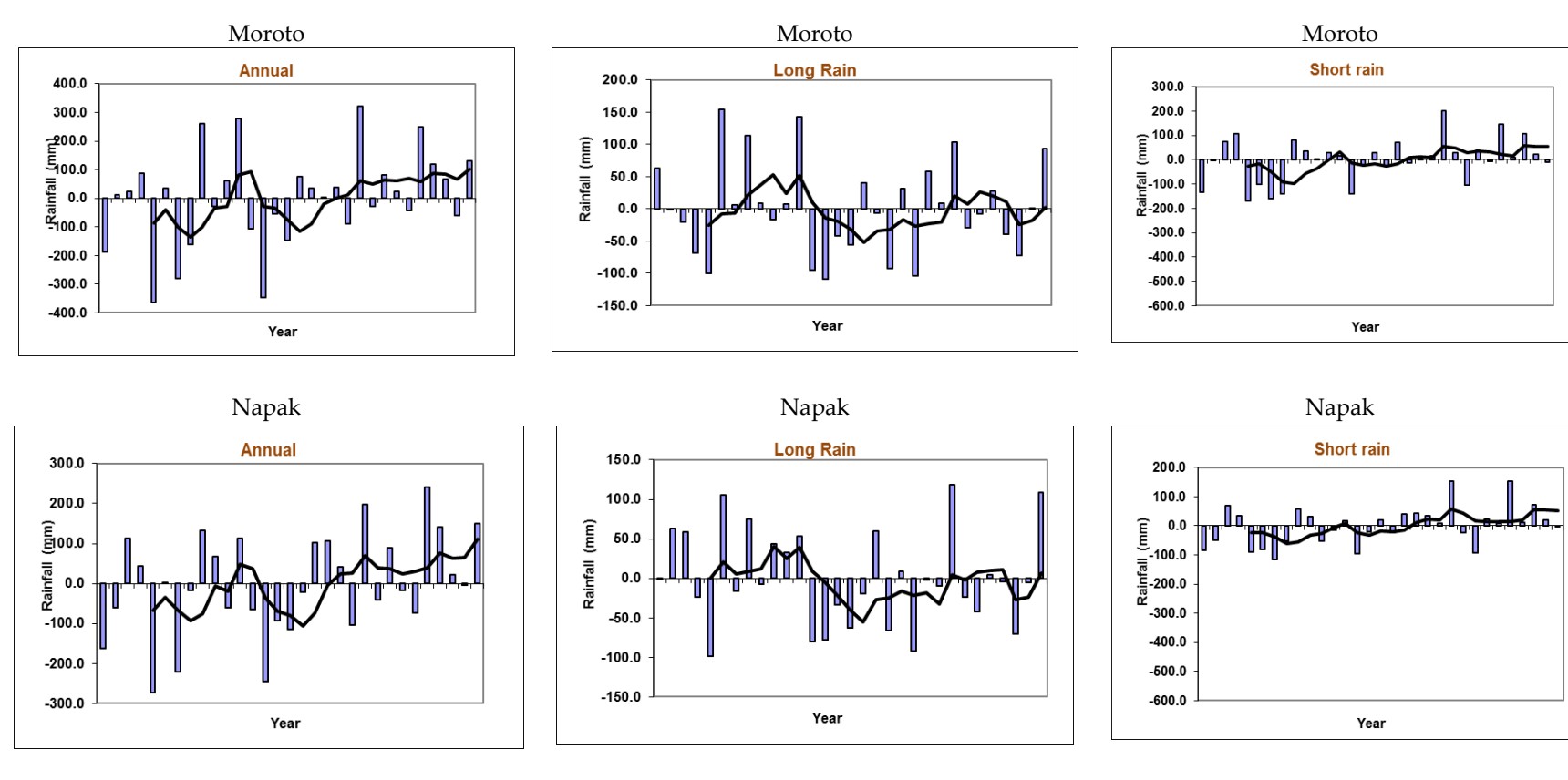

**Figure A1.** Historical rainfall.

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
