# Peer review of "Past, Present and Future Climate Trends Under Varied Representative Concentration Pathways for a Sub-Humid Region in Uganda"

_climate, doi:10.3390/cli7030035_

Round 1
Reviewer 1 Report
This manuscript describes a data trend analysis study that considered historical data, going back to the 1980s, for rainfall and temperature for the Karamoja sub-region of Uganda to look for climate change patterns. The study showed that minimum temperatures rose between 2000-2008 and a sharp rise in maximum temperature was seen in the early part of this century. The study suggested that the districts of Kaabong and Kotido are likely to see low rainfall levels in the future.
This is an interesting study that fits well within the journals scope and is likely to be of interest to a proportion of the journals readership but as is the case with localised studies, the main interest would come from Uganda.
The manuscript is quite well written and presented and the use of the English Language is good and to a standard suitable for publication. The introduction is informative and sets the scene well. It is also well referenced and the rationale for the work described. However, there are a few areas where the meaning is not quite clear. For example at line 109 the text states that climate data suffers from discontinuities. I presume the authors mean there are data gaps? Going on from the this at line 110 the text mentions civil unrest but it is not clear what that might have to do with data gaps. A few words of explanation would be beneficial.
I did find that the manuscript is very heavy on abbreviations and this does hinder the readability of the text. Would the authors please make an effort to keep abbreviations to a minimum. This is particularly a problem in sections 2.2, 2.4
At line 123 the text mentions ‘descriptive statistics’ – please give more detail.
Author Response
This manuscript describes a data trend analysis study that considered historical data, going back to the 1980s, for rainfall and temperature for the Karamoja sub-region of Uganda to look for climate change patterns. The study showed that minimum temperatures rose between 2000-2008 and a sharp rise in maximum temperature was seen in the early part of this century. The study suggested that the districts of Kaabong and Kotido are likely to see low rainfall levels in the future.
This is an interesting study that fits well within the journals scope and is likely to be of interest to a proportion of the journals readership but as is the case with localised studies, the main interest would come from Uganda.
· We appreciate this recognition towards the draft paper. It makes invaluable contribution at country level.
The manuscript is quite well written and presented and the use of the English Language is good and to a standard suitable for publication. The introduction is informative and sets the scene well. It is also well referenced and the rationale for the work described. However, there are a few areas where the meaning is not quite clear. For example at line 109 the text states that climate data suffers from discontinuities. I presume the authors mean there are data gaps? Going on from the this at line 110 the text mentions civil unrest but it is not clear what that might have to do with data gaps. A few words of explanation would be beneficial.
· An explanation to the effect that ground weather stations are often destroyed and where they are functional consistency in data collection is limited has been added into the text
I did find that the manuscript is very heavy on abbreviations and this does hinder the readability of the text. Would the authors please make an effort to keep abbreviations to a minimum. This is particularly a problem in sections 2.2, 2.4
· We appreciate this observation by the reviewer. We have however ensured that there is a full mention of the item before the abbreviation is put. This is because a couple of these abbreviations have become part and partial of the climate analysis literature. Many climate studies are abbreviation heavy to reduce repetitiveness of long sentences. See for example:
Dee, D. P., Uppala, S. M., Simmons, A. J., Berrisford, P., Poli, P., Kobayashi, S., ... & Bechtold, P. (2011). The ERA‐Interim reanalysis: Configuration and performance of the data assimilation system. Quarterly Journal of the royal meteorological society, 137(656), 553-597.
Blacutt, L. A., Herdies, D. L., de Gonçalves, L. G. G., Vila, D. A., & Andrade, M. (2015). Precipitation comparison for the CFSR, MERRA, TRMM3B42 and Combined Scheme datasets in Bolivia. Atmospheric Research, 163, 117-131.
At line 123 the text mentions ‘descriptive statistics’ – please give more detail.
We have made mention of the descriptive statistics to include means and standard deviations that were particularly used in this paper.

Reviewer 2 Report
I think the Editor to give me a chance to review this paper. I found some merits in the both methodology and results. However, I have some concerns on the different parts of the manuscript. If only the authors address carefully to all my comments, I will recommend publication of the manuscript.
Major comments
· The methods used are not well explained. The Mann-kendall test must be well presented in the methodology. Also I suggest that the student test should also be used to check if the average of the different projected sub-periods of rainfall and temperature are statistically significant different from the averages of the baseline period. This test can also allow them to check whether for each projected period, the average according to the rcp4.5 scenario is statistically different from the average of the rcp8.5 scenario. Se Alamou et al. (2017) and Obada et al. (2017) for statistic tests.
· Delta method also must be described. Is it a method of bias correction (delta change method) or something else?
· For the results, there is a total missing of the results of the various applications of Mann-kendall test.
· References are not ordered according to their appearance in the text.
· Appendix 1 allows to study the rainfall variability. This should allow you to clear wet periods and dry periods in the precipitation series. Please add the years in all the figures in the appendices.
· What do long rain and short rain mean?
Minor comments
Please find in the paper the minor comments.

Author Response
· The methods used are not well explained. The Mann-kendall test must be well presented in the methodology. Also I suggest that the student test should also be used to check if the average of the different projected sub-periods of rainfall and temperature are statistically significant different from the averages of the baseline period. This test can also allow them to check whether for each projected period, the average according to the rcp4.5 scenario is statistically different from the average of the rcp8.5 scenario. Se Alamou et al. (2017) and Obada et al. (2017) for statistic tests.
· We appreciate the observations by the reviewer. We have undertaken the following actions:
1. Ran the student test
2. Ran the Man-kendall test
· Delta method also must be described. Is it a method of bias correction (delta change method) or something else?
· We have explained the delta method approach. The delta method is a statistical analysis approach used in downscaling.
· For the results, there is a total missing of the results of the various applications of Mann-kendall test.
· We have observed the need to include the Mann-kendall test results. We have accordingly included the Mann-kendall test results in the process.
· References are not ordered according to their appearance in the text.
· We have re-ordered the references to ensure that it is starting from 1..Xn
· Appendix 1 allows to study the rainfall variability. This should allow you to clear wet periods and dry periods in the precipitation series. Please add the years in all the figures in the appendices.
· Thanks you very much for this. The graphs are system generated and the software does not input the years. We have maintained the graphics as generated by the system
· What do long rain and short rain mean?
· Long rains is the East African system is the same as wet season and the short rains is the same as the second rainy season just before the start of the dry period

Reviewer 3 Report
General comment
The manuscript by Anthony Egeru et al shows a welcome analysis of climate reanalyses and future projections for an understudied region in the world, namely the north of Uganda. However, the manuscript could be improved if the presentation of the climatology and the projections would be configured differently. Also, a number of misinterpretations of the climate change information need to be addressed and corrected. Finally, a number of apparent errors in the display of information are present that need to be corrected. As such, the manuscript is returned for major revision.
Presentation of the material
The analysis is based on reading 30 years of temperature and precipitation information from the MERRA reanalysis for a number of subregions in northern Uganda, and applying a delta-method to generate scenarios of future precipitation and temperature. The delta-factors are derived from 20 GCM projections by averaging results for 2 RCPs and 3 time slices. The resulting future time series are thus a combination of the MERRA reanalysis (more or less the “observed” state) and a mean climate change signal added to that observation. We see large variability in these time series that originate from the MERRA data, which is much larger than the climate change signal from the 20 averaged GCMs. Simultaneously, there is no information presented about how well the 20 GCMs agree on the (regional) climate change signal. My guess is that the difference between the 20 GCMs and the inherent natural variability will give a large uncertainty in the climate change signal. It would be much better to express this uncertainty in future climate by comparing the range in projections to the inter annual variability shown in the MERRA reanalysis. If the 20 GCMs large agree on a climate signal, the standard deviation of the 20 models will be small compared to the standard deviation of the 30 years in the reference climate. If the models strongly disagree, their standard deviation is much larger and the climate change information is not robust at all. This robustness is much more interesting than the display of the projected time series.
Misinterpretations of climate information
The authors point at a strong warming peak in 2017, and compare this peak to the observed global temperature data, in which 2017 was one of the highest values ever measured. However, your 2017 value is NOT a forecast, but just a repetition of the warm year 1989 that you took from MERRA, and to which you added the climate change delta’s. You cannot compare the individual years in your future time slices to observed records.
You construct a new 21st century time series by gluing the mid- and end-century time series (plus their climate change delta) together. On this time series you apply a trend analysis. This is a wrong interpretation of trends. Again, your projected time series are just repetitions of a historical time series. In the real world your temperature and precipitation record will be very different from how you constructed it. You can make a trend analysis by comparing mid- and end-century climate change values from the GCMs, but not by adding these signals on a repeated sequence of historical values.
Errors
In Figure 4 you use a label “Agago” which I cannot find in any table
In Table 1 and associated figures you show that max temperature is lower than mean temperature. I cannot understand how this can be true
From Gif 10 (and associated tables) it seems that for max temperature the difference between RCP4.5 and 8.5 tends to become smaller later in the century, while for mean and min temperature (figs 11 and 12) this difference tends to increase. Is this correct?
Minor comments
You start talking about drylands, but the region under study does not seem to be a drylands (700-1200 mm/yr)
You introduce MERRA and the 20 GCM projections without a lot of information about their quality or differences to other data sources. For instance, are MERRA results for the region of interest comparable to results from other data sets or reanalyses? How well do the climate models describe the current temperature and precipitation in the region of interest?
You introduce maximum and minimum temperature, but it is not clear what you mean. Is it the diurnal max/min, or the warmest/coldest day or month in a period?
Line 89-90: you state that there is more evaporation than precipitation. Where does the rest of the water come from?
In figure 1 you use the same legend entry for “Conservation areas” and “Karamoja sub-region”. That is confusing. Just mention that the region of interest is called Karamoja
Line 189-190: you state that end of century RCP8.5 would become dryer, but the projections still show a value larger than the current climate. It is much better to focus on the delta coming from the GCMs, and not on the absolute future rain/temperature values.
Line 221-222: you refer to marked inter annual variability, but all your information on inter annual variability comes from the MERRA data. Do the GCMs show a change in inter annual variability between early and late 21st century? That would be an interesting result
Figure 9.1-9.4 are quite nice, but it would be much better if (a) you would produce this result from MERRA first, and (b) show the same colours for contour intervals in each panel. This makes the spatial structure of the change in rainfall and temperature immediately clear
Table 5: “Sub-regional”, is that the same as “average over sub-regions”?
Line 295: wetting after 2000 is not shown clearly in your own figure 2
Line 483: typo in literature reference
Author Response
General comment
The manuscript by Anthony Egeru et al shows a welcome analysis of climate reanalyses and future projections for an understudied region in the world, namely the north of Uganda. However, the manuscript could be improved if the presentation of the climatology and the projections would be configured differently. Also, a number of misinterpretations of the climate change information need to be addressed and corrected. Finally, a number of apparent errors in the display of information are present that need to be corrected. As such, the manuscript is returned for major revision.
· We appreciate the time and technical input that the reviewer invested in providing coherent and in-depth review. The comments raised by the reviewer have not only helped to re-shape our thinking but also raised the need to invest more time to develop analytical capacities as well as infrastructure at institutional level to do such a job. That said, we have taken time to respond and address the comments raised by the review as much as possible that will be found satisfactory to merit publishing this article. The presentation of the figures and climatology information is premised on the purpose of the article to provide the first level information on projection for the Karamoja sub-region. We could have made it a complex presentation but this would only serve the interest of the highly technical persons in the field of climatology and lose the bulk of the potential users of this kind of information generated. This is the first article focusing on the Karamoja sub-region in terms of projection, thus achieving technical balance as well as providing opportunity for increased readership was critical to informing the structuring of this article.
Presentation of the material
The analysis is based on reading 30 years of temperature and precipitation information from the MERRA reanalysis for a number of subregions in northern Uganda, and applying a delta-method to generate scenarios of future precipitation and temperature. The delta-factors are derived from 20 GCM projections by averaging results for 2 RCPs and 3 time slices. The resulting future time series are thus a combination of the MERRA reanalysis (more or less the “observed” state) and a mean climate change signal added to that observation. We see large variability in these time series that originate from the MERRA data, which is much larger than the climate change signal from the 20 averaged GCMs. Simultaneously, there is no information presented about how well the 20 GCMs agree on the (regional) climate change signal. My guess is that the difference between the 20 GCMs and the inherent natural variability will give a large uncertainty in the climate change signal. It would be much better to express this uncertainty in future climate by comparing the range in projections to the inter annual variability shown in the MERRA reanalysis. If the 20 GCMs large agree on a climate signal, the standard deviation of the 20 models will be small compared to the standard deviation of the 30 years in the reference climate. If the models strongly disagree, their standard deviation is much larger and the climate change information is not robust at all. This robustness is much more interesting than the display of the projected time series.
· We do appreciate the observations that the reviewer has raised in this section regarding the presentation of material. We agree in part with most of the observations and have thus made some adjustments in terms of presentation whilst keeping in mind the review comments from other reviewers.
Misinterpretations of climate information
The authors point at a strong warming peak in 2017, and compare this peak to the observed global temperature data, in which 2017 was one of the highest values ever measured. However, your 2017 value is NOT a forecast, but just a repetition of the warm year 1989 that you took from MERRA, and to which you added the climate change delta’s. You cannot compare the individual years in your future time slices to observed records.
· Once again, we appreciate the comments raised by the reviewer. We have recoganised this fact and made revisions accordingly to ensure that it responds to the concern raised.
You construct a new 21st century time series by gluing the mid- and end-century time series (plus their climate change delta) together. On this time series you apply a trend analysis. This is a wrong interpretation of trends. Again, your projected time series are just repetitions of a historical time series. In the real world your temperature and precipitation record will be very different from how you constructed it. You can make a trend analysis by comparing mid- and end-century climate change values from the GCMs, but not by adding these signals on a repeated sequence of historical values.
· We have made significant corrections here as requested and deleted all the figures that raised concern and ensured that comparisons are made as indicated.
Errors
In Figure 4 you use a label “Agago” which I cannot find in any table
· This has been revised and corrected as Abim
In Table 1 and associated figures you show that max temperature is lower than mean temperature. I cannot understand how this can be true
· In normal climate data, this should not be a surprise since mean temperature is an average of maximum and minimum temperature. There is therefore no way the mean will be far higher than the maximum temperature. Rather, daily maximas are usually higher than the means. With means representing smoothed out values.
From Gif 10 (and associated tables) it seems that for max temperature the difference between RCP4.5 and 8.5 tends to become smaller later in the century, while for mean and min temperature (figs 11 and 12) this difference tends to increase. Is this correct?
· This is the correct picture. The issue is that the minimum temperature is rising faster than the maximum temperature.
Minor comments
You start talking about drylands, but the region under study does not seem to be a drylands (700-1200 mm/yr)
· Yes, in the Uganda context, this is a dryland and in-order to put it in the context of international drylands classification, it falls within the sub-humid drylands category, this is reflected in the sentence.
You introduce MERRA and the 20 GCM projections without a lot of information about their quality or differences to other data sources. For instance, are MERRA results for the region of interest comparable to results from other data sets or reanalyses? How well do the climate models describe the current temperature and precipitation in the region of interest?
· We take note of this. A comparability is described within the text with a correlation of 0.81 established. We have however, provided additional information on the model quality.
You introduce maximum and minimum temperature, but it is not clear what you mean. Is it the diurnal max/min, or the warmest/coldest day or month in a period?
· This is diurnal maximum and minimum, this because we are dealing with daily data.
Line 89-90: you state that there is more evaporation than precipitation. Where does the rest of the water come from?
· This has been amended to read “The region has a high evapotranspiration”
In figure 1 you use the same legend entry for “Conservation areas” and “Karamoja sub-region”. That is confusing. Just mention that the region of interest is called Karamoja
· A new Figure 1 has been developed as this was also requested by Reviewer 1
Line 189-190: you state that end of century RCP8.5 would become dryer, but the projections still show a value larger than the current climate. It is much better to focus on the delta coming from the GCMs, and not on the absolute future rain/temperature values.
· This has been revised as the reviewer 2 had pointed out that this was a mix of discussion; to which was true. It was an attempt at interpretation of result but has since then been revised accordingly-deleted!
Line 221-222: you refer to marked inter annual variability, but all your information on inter annual variability comes from the MERRA data. Do the GCMs show a change in inter annual variability between early and late 21st century? That would be an interesting result
· We have taken this into consideration and revised accordingly.
Figure 9.1-9.4 are quite nice, but it would be much better if (a) you would produce this result from MERRA first, and (b) show the same colours for contour intervals in each panel. This makes the spatial structure of the change in rainfall and temperature immediately clear
· We prefer that it is represented as currently presented using the IDW in the ArcGIS environment and the numbering is now revised following the removal of other figures.
Table 5: “Sub-regional”, is that the same as “average over sub-regions”?
· This has been corrected to sub-region’s average
Line 295: wetting after 2000 is not shown clearly in your own figure 2
· This statement is not referenced to Figure 2 but Figures in Annex 1 that show Annual rainfall, first season rains-wet season (long rains) and second season rains (short rains).
Line 483: typo in literature reference
· This has been corrected

Round 2
Reviewer 2 Report
The authors took into account some of my major comments. But unfortunately, others remained unfulfilled. The student text, the equations of the delta change method have not been taken into account. But this does not diminish the quality of the work. I recommend publication of manuscript after taking into account some minor errors that persist in the paper. For example it is necessary to harmonize the writing of the statistic Z (Z or Zs), in the equation 5 it is necessary to add space between if and S.
It is also necessary to check the values of Z because for a significance trend at 95% confidence level, | Z | must be greater than 1.96 for a sample size about 30.
Author Response
The authors took into account some of my major comments. But unfortunately, others remained unfulfilled. The student text, the equations of the delta change method have not been taken into account. But this does not diminish the quality of the work. I recommend publication of manuscript after taking into account some minor errors that persist in the paper. For example it is necessary to harmonize the writing of the statistic Z (Z or Zs), in the equation 5 it is necessary to add space between if and S.
It is also necessary to check the values of Z because for a significance trend at 95% confidence level, | Z | must be greater than 1.96 for a sample size about 30.
· We have harmonized the Z (Z or Zs), as Zs, by checking the equation set-up the ‘s’ had been missed while writing it out. We have rectified this. We have also generated afresh the statistics that are in position to show that | Z | must be greater than 1.96 for a sample size about 30 at 95% confidence level. We did this by changing the software from the XLSTAT to the MAKESENS-application for trend calculation provided by the Finish Meteorological Institute. It appears the XLSTAT has some issues with the computations and we shall also bring this to the attention of the developers so that other researchers do not get into similar challenges. We appreciate the reviewer’s intervention in this regard.
